# A Spam Filtering Method Based on Multi-Modal Fusion

**Hong Yang** [†] [ID]**, Qihe Liu** *,[†]**, Shijie Zhou** [†] **and Yang Luo** [†]

The School of Information and Software Enginerring, University of Electronic Science and Technology of China, Chengdu 610054, China; yh@std.uestc.edu.cn (H.Y.); sjzhou@uestc.edu.cn (S.Z.); luoyanum@163.com (Y.L.)

* Correspondence: qiheliu@uestc.edu.cn; Tel.: +86-152-0829-2978

† Current address: No. 4, Section 2, Jianshe North Road, Chenghua District, Chengdu 610054, China.

**Abstract:** In recent years, the single-modal spam filtering systems have had a high detection rate for image spamming or text spamming. To avoid detection based on the single-modal spam filtering systems, spammers inject junk information into the multi-modality part of an email and combine them to reduce the recognition rate of the single-modal spam filtering systems, thereby implementing the purpose of evading detection. In view of this situation, a new model called multi-modal architecture based on model fusion (MMA-MF) is proposed, which use a multi-modal fusion method to ensure it could effectively filter spam whether it is hidden in the text or in the image. The model fuses a Convolutional Neural Network (CNN) model and a Long Short-Term Memory (LSTM) model to filter spam. Using the LSTM model and the CNN model to process the text and image parts of an email separately to obtain two classification probability values, then the two classification probability values are incorporated into a fusion model to identify whether the email is spam or not. For the hyperparameters of the MMA-MF model, we use a grid search optimization method to get the most suitable hyperparameters for it, and employ a k-fold cross-validation method to evaluate the performance of this model. Our experimental results show that this model is superior to the traditional spam filtering systems and can achieve accuracies in the range of 92.64–98.48%.

**Keywords:** spam filtering system; multi-modal; MMA-MF; fusion model; LSTM; CNN

---

## 1. Introduction

Spam can be defined as an email which contains unsolicited mail [1]. With the rapid development of the Internet, Internet users are increasingly using emails to communicate. At the same time, the issue of spam is getting worse, in which the purpose of most spam is to solicit the recipients for money. In order to achieve this, the products they provide claim to miraculously cure health problems such as diabetes, obesity and hair loss. They may be of any nature, whether it is an advertisement, a text email, an image email or a email that contains text and image data. According to the spam analysis report of Kaspersky Lab, a well-known organization in the security field, the average proportion of global spam in total emails were as high as 56.63% or more in 2017 [2]. This phenomenon indicates that spam is flooding the entire network, which brings inconvenience to cyber citizens. For text spam or image spam, the single-modal spam filtering systems have a high detection rate, while, in order to escape detection, spammers may insert junk information into the multi-modal part of an email, which we call it hybrid spam, to reduce the detection rate of the single-modal spam filtering systems, ultimately achieving the purpose of evading detection. For hybrid spam, it is more harmful than traditional spam because it contains more information than traditional spam, and it requires more network bandwidth and storage space for forwarding and delivery of the mailbox servers. Moreover, viruses or unsolicited information carried by hybrid spam are more difficult to detect, which brings tremendous information security risks to people's communication. Therefore, it is extremely important to learn how to effectively identify hybrid spam.

---

In machine learning and cybersecurity communities, anti-spam methods have been studied for many years [3–15]. These methods roughly are classified into three categories: (1) text-based spam detection; (2) image-based spam detection; and (3) multi-modal spam detection. The first and second categories primarily use the textual content or image content of an email to filter spam, respectively. However, the last category processes both the textual and image content of an email to filter spam.

For text-based spam detection, the specific research contents are as follows: Carreras [3] uses the decision tree method to filter spam, which uses the RLM (R. LOPEZ DE MANTARAS) distance method instead of the information gain method to select text features. The experimental results show that the accuracy and recall rate of this method are above 88%. Androutsopoulos [4] employs multiple weak classifiers to get multiple classification probability values of an email as spam, thereinto every weak classifier uses a logarithmic regression method to obtain the classification probability value. Eventually, using a boosting method to combine the multiple classification probability values to get a real value of the email as spam, which is compared to a threshold to determine whether the email is spam or not. The remaining methods use other methods to filter spam, such as Bayesian algorithm [5–8], k-Nearest Neighbor (k-NN) algorithm [9] and deep learning algorithm [10]. These methods have good performance for text spam detection. However, if most of the junk information is included in the image section, and the text section does not contain the junk information or only contains a small amount of it, will cause the spam filtering systems to treat it as a normal email.

For image-based spam detection, some methods have also been proposed and demonstrated good performance. Er-Xin Shang [12] proposed a method to use the convolutional neural network to automatically extract image features and detect spam. Wang [13] proposed a method that uses a multi-model combination method to identify image spam. Kumar [14] proposed an supported vector machine (SVM) method based on Gaussian kernel to filter spam. Congfu [15] proposed a multi-feature combination method to design a spam filtering system, it extracts the texture features and attribute features of the image part in an email separately, and then classifies them by the SVM method to get two classification probability values, finally entering them into the model designed by the SVM method again to get a real value of the email as spam. These methods can deal with image spam, while it can't handle text spam and hybrid spam.

For multi-modal spam detection, Yang [16] proposed a method based on multi-modal feature fusion. The main idea is to use the P-SVM method to construct multi-classifiers for the text and image data in an email respectively, and then adopts the SVM method to fuse the output of the multi-classifier to get a true value of the email as spam. This method can handle with text-based spam, image-based spam and hybrid spam, but the experimental results show that the identifying accuracy of spam is only 90%. Recently, deep learning techniques have been widely applied and obtained huge success in many application domains. Particularly, the convolutional neural network (CNN) has higher accuracy for image classification tasks than other methods [17], and the Long Short-Term Memory (LSTM) neural network has been widely used in natural language processing due to its time and memory characteristics [18]. It inspires us to adopt the CNN and LSTM model to deal with the image and text data for hybrid emails, respectively.

Keeping the research line based on multi-modal spam detection, in this paper, we design a multi-modal fusion model for spam detection, which called MMA-MF. For the text part of an email, we use an LSTM model to extract textual semantic relations features and obtain classification probability value of the text part as spam. For the image part of the same email, we construct an CNN model to get classification probability value of the image part as spam. From the two models, we acquire two classification probability values. Then, the two classification probability values are fed into a fusion model designed by the logistic regression method to acquire the eventual classification probability value, which describes the true value of the email as spam. If an email just only has text data or image data, we use dropout cogitation [19] to set the classification probability value of the LSTM model output or the CNN model output become 0.5 to ensure that the MMF-MA model could not only handle text spam and image spam, but also handle hybrid spam. In addition, the grid search optimization

algorithm [20] is used to select the training hyperparameters of the model, and k-fold cross-validation method [21] is employed to verify the MMA-MF model performance.

The proposed method merges the text and image information in an email, so it can efficiently filter spam whether the junk information is hidden in the text or in the image. That is, the advantage of the MMA-MF model is that it can not only filter hybrid spam, but also filter spam with only text data or image data. The experimental results indicate that our method is better than other methods significantly. The main contribution is that we apply the CNN and LSTM model to handle the image and text data in an email, and combine them into a fusion model by the logistic regression method. To our best knowledge, we firstly shed light on this approach in the email filtering systems.

The rest of this paper is organized as follows: Section 2 describes the architecture of the MMA-MF model, we present the design framework of the CNN, LSTM and fusion model, the brief categorization algorithm for text spam and image spam. Section 3 presents evaluation metrics and validation schemes. Section 4 is about experimental results and discussion. In the end, conclusions are given in Section 5.

## 2. MMA-MF Model Architectures

Essentially, the spam filtering system is a binary classification problem. In order to make our model not only filter hybrid spam but also filter spam with only text data or image data, we propose a kind of spam filtering framework called MMA-MF. This framework shows in Figure 1.

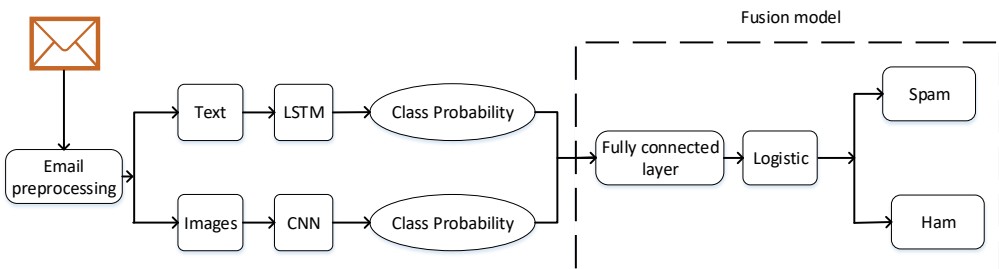

**Figure 1.** MMA-MF model architecture.

The specific steps of the MMA-MF model to identify spam are described as follows:

1. Email preprocessing: separate the text and image data from an email to obtain the text dataset and the image dataset.
2. Obtaining the optimal classifiers: the text dataset and the image dataset are used to train and optimize the LSTM model and the CNN model, respectively—finally getting the optimal LSTM model and the optimal CNN model.
3. Obtaining the classification probability values: the image dataset is re-entered into the optimal CNN model to obtain the classification probability values of the image dataset as spam. Similarly, the text dataset is re-entered into the optimal LSTM model to obtain the classification probability values of the text dataset as spam. For an email that only has text data or image data, we use dropout ideology to set the corresponding model output probability value $p = 0.5$.
4. Obtaining the optimal fusion model: the two classification probability values are fed into the fusion model to train and optimize it, ultimately getting the optimal fusion model.

In the above descriptions, through by steps 1, 3 and 4, we can get the classification probability value of a new email as spam, whether the new email is a hybrid email or a single-modal email. In conclusion, we give the overall framework of the MMF-MA model and the brief steps for obtaining the classification probability value of an email as spam. Next, we will introduce the internal structure of the LSTM model, the CNN model and the fusion model, and the selection of the optimal hyperparameter values for the three models in detail.

*2.1. Text Classification Model: LSTM Model*

The structure of the LSTM model is roughly shown in Figure 2. It is composed of a one word embedded layer, two LSTM layers and one fully connected (FC) layer. The steps of handling the text portion of an email to obtain the classification probability value of the email are as follows: firstly using the preprocessing technique to acquire the text data of an email, then using the word embedding technique to get its word vector representation. In this paper, we select the word2vec toolkit to get word vector representation. After that, we use the designed two LSTM layers to automatically extract features from the text data. Finally, we apply the FC layer with Softmax activation function to obtain the classification probability value of the text data as spam, and the LSTM model is trained and optimized by using the log-likelihood function to minimize the loss function [22].

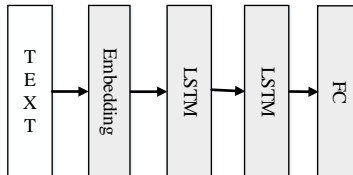

**Figure 2.** LSTM model framework.

For the hyperparameters of the LSTM model, we use the grid search optimization algorithm to select the optimal values for the five hyperparameters, which are learning rate, batch size, epochs, dropout rate and optimization algorithm. The range and optimal values of these hyperparameters selected by the LSTM model are shown in Table 1.

**Table 1.** The range and optimal values of hyperparameters for LSTM.

| Hyperparameter | Range | Optimal Value |
| --- | --- | --- |
| learning rate | [0.001, 0.01, 0.1, 0.2] | 0.001 |
| batch size | [8, 16, 32] | 32 |
| epochs | [10, 20, 30] | 30 |
| dropout rate | [0.2, 0.3, 0.4] | 0.3 |
| optimization algorithm | [SGD [23], RMSprop [24], Adam [25]] | Adam |

We make a brief pseudo code description here for the LSTM model. For a detailed algorithm about the LSTM unit, please see the literature [10,26]. Let $T$ denote the text data of an email. Input $T$ into the embedding step to convert $T$ into becoming a word vector $x$, $x = (x_1, x_2, \cdots, x_l)$, where $x_i \in R^n$ is the n-dimensional word vectors for the $i$-th word in the document $T$ and matrix $x \in R^{l \times n}$ denote the document $T$, where $l$ is the max length of and $l \leq 500$. At time-step $t$, the memory $c_t$ and the hidden state $h_t$ are updated with the following equations:

$$\begin{bmatrix} i_t \\ f_t \\ o_t \\ \hat{c}_t \end{bmatrix} = \begin{bmatrix} \sigma \\ \sigma \\ \sigma \\ tanh \end{bmatrix} W \cdot [h_{t-1}, x_t], \tag{1}$$

$$c_t = f_t \odot c_{t-1} + i_t \odot \hat{c}_t, \tag{2}$$

$$h_t = o_t \odot tanh(c_t), \tag{3}$$

where $x_t$ is the input at the current time-step, $i$, $f$ and $o$ is the input gate activation, forget gate activation and output gate activation, respectively, $\hat{c}_t$ is the current cell state, $\sigma$ denotes the logistic sigmoid function and $\odot$ denotes element-wise multiplication. Through training and optimizing the

LSTM model, we could obtain the classification probability value of the text part as spam. The entire process of text spam classification algorithm is described in Algorithm 1.

---

**Algorithm 1** Text Spam Classification Algorithm.

---

**Input:** Text Document T
**Output:** Text spam classification probability value $e$
  1: Input $T$ into the word2vec toolkit to get the word vector $x$, $x = (x_1, x_2, \cdots, x_l)$.
  2: For the first LSTM layer (64 LSTM units), input $x$ at time $t$ and complete the following calculations:

$$\begin{bmatrix} i_t \\ f_t \\ o_t \\ \hat{c}_t \end{bmatrix} = \begin{bmatrix} \sigma \\ \sigma \\ \sigma \\ tanh \end{bmatrix} W \cdot [h_{t-1}, x_t],$$

$$c_t = f_t \odot c_{t-1} + i_t \odot \hat{c}_t,$$

$$h_t = o_t \odot tanh(c_t).$$

  3: By the first LSTM layer, getting the text feature vector $h = (h_1, h_2, \cdots, h_{64})$.
  4: For the second LSTM layer(32 LSTM units), input $h$ at time t and do the same as Equations (1)–(3).
     Finally, getting more abstract text feature vector $k$, $k = (k_1, k_2, \cdots, k_{32})$.
  5: Input $k$ to FC layer and using Softmax activation function to gain the text classification probability
     value $e$;
  6: **return** $e$;

---

The sequences of input (sentences) are fed into the LSTM unit along with the output of the previous LSTM unit. This is repeated with each input sentence and in this way the LSTM units keep on saving the important features. The number of LSTM units save the most important features. Hence, through the LSTM layer, FC layer and Softmax activation function, we can gain the classification probability value $e$ of the text part as spam.

### 2.2. Image Classification Model: CNN Model

In this subsection, we design a CNN model to classify an email. For the hyperparameters of the CNN model, we also use the grid search optimization algorithm to select the optimal values for the four hyperparameters, which are learning rate, batch size, epochs and optimization algorithm. The range and optimal values of these hyperparameters selected by the CNN model are shown in Table 2.

**Table 2.** The range and optimal values of hyperparameters for CNN.

| Hyperparameter | Range | Optimal Value |
| --- | --- | --- |
| learning rate | [0.001, 0.01, 0.1, 0.2] | 0.01 |
| batch size | [5, 10, 20] | 20 |
| epochs | [8, 16, 32] | 32 |
| optimization algorithm | [SGD, RMSprop, Adam] | SGD |

For an image email, it is scaled to $128 \times 128$ RGB size, and then input it to the CNN model to obtain the classification probability value of the image part as spam. This model contains three convolutional layers, in which the first convolutional layer contains 32 filters, the second and third convolutional layer contain the same number of filters, 64. The filter kernel size of each convolutional layer is $5 \times 5$, and the convolution stride and space padding are fixed to one pixel. Each convolutional

layer is connected to a 2 × 2 window of MaxPooling, with a stride of one. After that, we adopt flatten technology to make a multidimensional vector into a one-dimensional vector. Finally, using three fully connected layers with 64, 32, and 2 neurons to obtain the classification probability value of the image data as spam. Neurons in all hidden layers are followed by ReLu nonlinear function as activation function. Furthermore, batch normalization technology is adopted to normalize the image dataset after each convolutional layer to prevent the image dataset distribution from changing during the training process, thereby avoiding gradient disappearance or explosion, and accelerating the CNN model training by reducing internal covariate shift [27]. The brief description of the CNN model architecture, the output shapes and parameters for each layer are described in Table 3.

**Table 3.** CNN model architecture description.

| Layer (Type) | Output Shape | Param |
|---|---|---|
| conv2d_1 (Conv2D) | (None, 128, 128, 32) | 2432 |
| batch_normalization_1 | (Batch (None, 128, 128, 32) | 128 |
| activation_1 (Activation) | (None, 128, 128, 32) | 0 |
| max_pooling2d_1 | (MaxPooling2 (None, 64, 64, 32) | 0 |
| conv2d_2 (Conv2D) | (None, 64, 64, 64) | 51264 |
| batch_normalization_2 | (Batch (None, 64, 64, 64) | 256 |
| activation_2 (Activation) | (None, 64, 64, 64) | 0 |
| max_pooling2d_2 | (MaxPooling2 (None, 32, 32, 64) | 0 |
| conv2d_3 (Conv2D) | (None, 32, 32, 64) | 102464 |
| batch_normalization_3 | (Batch (None, 32, 32, 64) | 256 |
| activation_3 (Activation) | (None, 32, 32, 64) | 0 |
| max_pooling2d_3 | (MaxPooling2 (None, 16, 16, 64) | 0 |
| flatten_1 (Flatten) | (None, 16384) | 0 |
| dense_1 (Dense) | (None, 64) | 1048640 |
| batch_normalization_4 | (Batch (None, 64) | 256 |
| activation_4 (Activation) | (None, 64) | 0 |
| dense_2 (Dense) | (None, 32) | 2080 |
| dense_3 (Dense) | (None, 2) | 66 |

The CNN algorithm details have been described in Ref. [28]; therefore, we only give a brief description for the CNN model. The whole process of image spam classification algorithm is described in Algorithm 2. Taking image $m$ as an input to the CNN model, the classification probability value $g$ of the image part as spam is ultimately obtained.

---

**Algorithm 2** Image Spam Classification Algorithm.

---

**Input:** Image $m$, size 128 × 128 RGB
**Output:** Image spam classification probability value $g$
  1: Input $m$ to the 3 convolutional layers, getting the results $d$, $d = (d_1, d_2, \cdots, d_{64})$;
  2: Input $d$ to the first 2 FC layers, which contain 64 and 32 neurons, gaining feature vector $c$,
     $c = (c_1, c_2, \cdots, c_{32})$;
  3: Input $c$ to the last FC layer, which contains two neurons, using a Softmax activation function to
     obtain classification probability value $g$;
  4: **return** $g$;

---

### 2.3. Fusion Model

The structure of the fusion model is shown in Figure 3. The aim is to fuse the classification probability value of an email text part with the classification probability value of the same email image

part to obtain the most accurate classification probability value of the email as spam. The overall steps are as follows: 1. Combining the two classification probability values of the LSTM and CNN models to get a feature vector $q$, $q \in R^{1 \times 4}$; 2. Inputting $q$ into the FC layer with 64 neurons to get a comprehensive feature vector; 3. Inputting the comprehensive feature vector to the logistic layer, which includes two neurons and chooses the logistic regression function as the activation function to get the most accurate classification probability value of the email as spam. Taking into account the efficiency of our machine, we only use the grid search optimization algorithm to select the optimal values for the four hyperparameters, which are learning rate, batch size, epochs and optimization algorithm, the best hyperparameter for learning rate is equal to 0.01, batch size is equal to 16, epochs is equal to 30 and the optimization algorithm is the SGD algorithm.

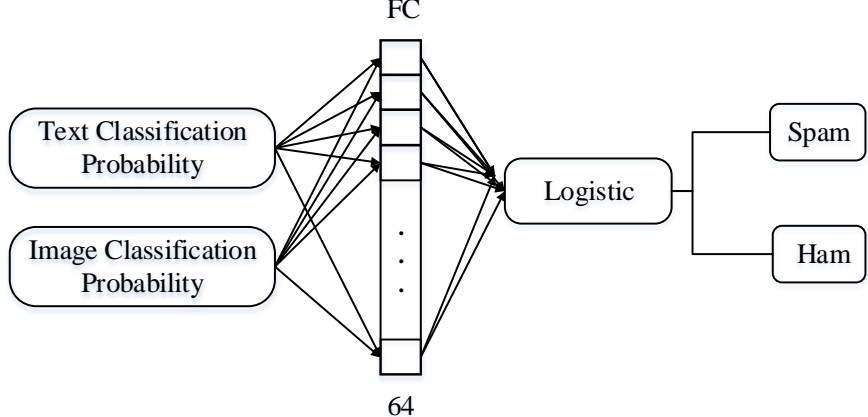

**Figure 3.** Fusion model structure.

Suppose that the classification probability dataset input to the fusion model is $D=\{(q_1,y_1), (q_2,y_2), \cdots, (q_v,y_v)\}$, $q_i \in R^{1 \times 4}$, $y_i \in \{0,1\}$, in which the conditional probability distribution of the logistic regression function is as follows:

$$P(Y = 1|q) = \pi(q) = \frac{e^{-w^T \cdot q}}{1 + e^{-w^T \cdot q}}, \tag{4}$$

$$P(Y = 0|q) = 1 - \pi(q) = \frac{1}{1 + e^{-w^T \cdot q}}. \tag{5}$$

We choose the log-likelihood function as the loss function, and the formula is as follows:

$$\begin{aligned} L(w) &= \sum_{i=1}^{v} [y_i \log \pi(q_i) + (1 - y_i) \log(1 - \pi(q_i))] \\ &= \sum_{i=1}^{v} \left[ y_i \log \frac{\pi(q_i)}{1 - \pi(q_i)} + \log(1 - \pi(q_i)) \right] \\ &= \sum_{i=1}^{v} [y_i(w \cdot q_i) - \log(1 + e^{(w \cdot q_i)})]. \end{aligned} \tag{6}$$

The maximum value of $L(w)$ is obtained by the Adam algorithm. In addition, the optimal estimate value of the parameter $w$ can be obtained by optimizing $L(w)$. If $p > 0.5$, it means that the email is spam; otherwise, it is a normal email.

## 3. Evaluation Metrics and Validation Scheme

### 3.1. Evaluation Metrics

In order to assess the effectiveness of the proposed method, different evaluation indicators have been used, including accuracy, recall, precision and f1-score, which are defined as follows:

$$Accuracy = \frac{TP + TN}{TP + TN + FP + FN}, \tag{7}$$

$$Recall = \frac{TP}{TP + FN}, \tag{8}$$

$$Precision = \frac{TP}{TP + FP}, \tag{9}$$

$$F1 - Score = \frac{2 * (Precision * Recall)}{Precision + Recall}. \tag{10}$$

The specific meanings of FP, FN, TP and TN are defined as follows:

- False Positive (FP): The number of legitimate emails (Ham) that are misclassified;
- False Negative (FN): The number of misclassified spam;
- True Positive (TP): The number of spam that are correctly classified;
- True Negative (TN): The number of legitimate emails (Ham) that are correctly classified.

For spam detection, the evaluation metrics about accuracy, recall, precision and f1-score are mainly based on the confusion matrix, which shows in Table 4:

**Table 4.** Confusion matrix.

| Prediction | Actual | |
|---|---|---|
| | Spam | Ham |
| Spam | TP | FN |
| Ham | FP | TN |

### 3.2. Validation Scheme

In previous studies, a rejection verification scheme has been employed to evaluate the effectiveness of the built spam filtering system. Different studies use different training-test split percentages for data distribution, in which the training dataset is used to evaluate the performance of a model; the testing dataset is used to obtain the accuracy of the selected optimal model. The easiest and most straightforward way is to divide the dataset into two parts, one for training and the other for testing, which is called the hold out method. The shortcoming is that the evaluation depends largely on which samples end up in which collection. Another way to reduce the variance of the hold out method is the k-fold cross-validation method, in the k-fold cross-validation method, the dataset $M$ is divided into k mutually exclusive parts, and $M_1, M_2, \cdots, M_k$. The inducer is trained on $M_i/M$ and tested against $M_i$. This is repeated $k$ times with different $i$, $i = 1, 2, \cdots, k$. For a k-fold test, the accuracy, recall, precision and f1-score are defined as follows:

$$Accuracy = \sum_{i=1}^{k} Accuracy_i, \tag{11}$$

$$Recall = \sum_{i=1}^{k} Recall_i, \tag{12}$$

$$Precision = \sum_{i=1}^{k} Precision_i, \tag{13}$$

$$F1 - Score = \sum_{i=1}^{k} F1 - Score_i, \tag{14}$$

where $Accuracy_i$, $Recall_i$, $Precision_i$ and $F1 - Score_i$ are the accuracy, recall, precision and f1-score for each of the k tests. Considering the performance of our computer, we choose a 5-fold cross-validation method throughout the experiments.

## 4. Experimental Results and Discussion

### 4.1. Corpus

In this paper, we choose three types of email datasets for our experiments: the dataset only contained text, the dataset only contained image and the mixed dataset that contains image and text data. The dataset only containing text comes from the Enron corpus [29], and we only choose 6000 text emails (4500 Spam, 1500 Ham) by removing duplicates and randomly selecting from 33,645 text emails. The dataset only containing images is composed of Personal Image Ham, Personal Image Spam and Spam Archive Image Spam [30] after culling the data with a picture size is equal to 0 and simply getting rid of the duplication images. Since there is no public mixed dataset that contains both image and text data, in order to get the mixed dataset, we choose to combine the Enron dataset with Personal Image Ham, Personal Image Spam and Spam Archive Image Spam, eliminating invalid data and randomly selecting them to form the mixed dataset 1 and 2. The dataset details used in the experiments are shown in Table 5 below.

**Table 5.** Datasets used in experiments.

| Type | Original Dataset | Before Remove Duplicates | After Remove Duplicates |
|---|---|---|---|
| Text | Enron Ham | 17,108 | 1500 |
| | Enron Spam | 16,537 | 4500 |
| Image | Personal Image Ham | 2021 | 1393 |
| | Personal Image Spam | 3298 | 130 |
| | Spam Archive Image Spam | 16,031 | 1263 |
| Mixed Dataset 1 | Ham:Enron&Personal Image | 19,129 | (Text:Image) 600:600 |
| | Spam:Enron&Personal Image&Archive Image | 35,866 | (Text:Image) 600:600 |
| Mixed Dataset 2 | Ham:Enron&Personal Image | 19,129 | (Text:Image) 600:300 |
| | Spam:Enron&Personal Image&Archive Image | 35,866 | (Text:Image) 600:300 |

For the mixed dataset 1, the number of text dataset is equal to the number of image dataset, which contains 600 Spam (text Spam 600 and Image Spam 600 are formed into 600 spam) and 600 Ham (text Ham 600 and Image Ham 600 are formed into 600 Ham email). For the mixed dataset 2, the number of text dataset is not equal to the number of image dataset, which contains 600 Spam (through by dropout technology, text Spam 600 and Image Spam 300 are formed into 600 Spam) and 600 Ham (through by dropout technology, text Ham 600 and Image Ham 300 are formed into 600 Ham). We selected 5-fold cross-validation method to evaluate the MMA-MF model performance. From the principle of k-fold cross-validation method introduced in Section 3.2, we can derive the size of the training and testing datasets for the four datasets that shows in Table 6. Since each text email and image email are only mixed once to get one mixed email, this ensures that the training and test datasets do not overlap at all.

**Table 6.** Training and testing dataset size.

| Type | Training Dataset Size | Testing Dataset Size |
|---|---|---|
| Text | 5000 | 1000 |
| Image | 2322 | 464 |
| Mixed Dataset 1 Mixed Dataset 2 | 960 | 240 |

### 4.2. Results and Discussion

In this section, we show our evaluation results on image spam classification, text spam classification and the mixed spam classification. Moreover, we give some analysis and discussions for the experimental results.

We use 5-fold cross-validation method to verify the performance of the MMA-MF model on the text dataset, image dataset and the mixed datasets 1 and 2, and obtain the experimental results of the MMA-MF model on the four datasets, as shown in Table 7, in which $\bar{u}$ means the average value of Accuracy, Recall, F1-Score or Precision after using the 5-fold cross-validation method.

**Table 7.** Experimental results in 5-fold cross-validation for the MMA-MF model.

| Fold | Accuracy | Recall | F1-Score | Precision |
|---|---|---|---|---|
| MMA-MF Model for Text Dataset | | | | |
| 1 | 0.9842 | 0.9784 | 0.9724 | 0.9850 |
| 2 | 0.9867 | 0.9815 | 0.9747 | 0.9850 |
| 3 | 0.9867 | 0.9819 | 0.9765 | 0.9900 |
| 4 | 0.9825 | 0.9771 | 0.9727 | 0.9800 |
| 5 | 0.9842 | 0.9789 | 0.9753 | 0.9850 |
| $\bar{u}$ | 0.9848 | 0.9796 | 0.9743 | 0.9850 |
| MMA-MF Model for Image Dataset | | | | |
| 1 | 0.9335 | 0.9264 | 0.9289 | 0.9050 |
| 2 | 0.9256 | 0.9263 | 0.9275 | 0.9001 |
| 3 | 0.9150 | 0.9233 | 0.9183 | 0.9350 |
| 4 | 0.9235 | 0.9283 | 0.9297 | 0.9200 |
| 5 | 0.9344 | 0.9272 | 0.9271 | 0.9250 |
| $\bar{u}$ | 0.9264 | 0.9263 | 0.9263 | 0.9170 |
| MMA-MF Model for Hybrid Dataset 1 | | | | |
| 1 | 0.9792 | 0.9792 | 0.9792 | 0.9750 |
| 2 | 0.9917 | 0.9920 | 0.9917 | 0.9900 |
| 3 | 0.9750 | 0.9758 | 0.9750 | 0.9750 |
| 4 | 0.9917 | 0.9911 | 0.9916 | 0.9900 |
| 5 | 0.9792 | 0.9787 | 0.9791 | 0.9800 |
| $\bar{u}$ | 0.9833 | 0.9834 | 0.9833 | 0.9820 |
| MMA-MF Model for Hybrid Dataset 2 | | | | |
| 1 | 0.9875 | 0.9877 | 0.9875 | 0.9850 |
| 2 | 0.9875 | 0.9867 | 0.9874 | 0.9900 |
| 3 | 0.9833 | 0.9846 | 0.9833 | 0.9800 |
| 4 | 0.9896 | 0.9913 | 0.9895 | 0.9850 |
| 5 | 0.9750 | 0.9759 | 0.9750 | 0.9750 |
| $\bar{u}$ | 0.9846 | 0.9852 | 0.9845 | 0.9830 |

From Table 7, we can conclude that the MMA-MF model designed in this paper implements the filtering function of spam, whether it is hidden in the text, in the image, or hidden in the text and image, we are all able to handle it and filter it out pretty well. In conclusion, we have the following observations: for the MMA-MF model, it not only filters well mixed emails, but also filters text emails or image-based emails well.

In order to further verify the performance of the MMA-MF model, we compare it with a large set of well-performed models by using the four datasets. For the text dataset, we use Character-level CNN (Char-CNN) [31], BiLSTM (Bi-directional LSTM) [31], Naive Bayes, the Immune Cross-Regulation Model (ICRM) [11], and the MMA-MF model for comparison. For the image dataset, we use Naive Bayes, ID3 Decision Tree and the MMA-MF model for comparison. For the mixed dataset 1 and 2, we use Nguyen's idea of incorporating the k-NN algorithm into a model [32] to change the fusion model part of the MMA-MF model. Besides using the fusion model designed by the MMA-MF model, we also use a classical k-nearest neighbors (classical k-NN) model, an enhanced k-nearest neighbors (enhanced k-NN) model, which assigns email labels to each email based on the centroid displacement of the classes present in the neighboring points, a SVM model and a random forest model to replace the fusion model designed by the MMA-MF model to prove that the fusion model designed by the MMA-MF model is better than other models. At the same time, the hyperparameters for the classical k-NN, enhanced k-NN, SVM and random forest models are also using the grid search optimization algorithm to select the best hyperparameters for the four models. For instance, through the grid search optimization algorithm, the best hyperparameters C and gamma of the SVM model are equal to 1 and 0.001, respectively, and the best hyperparameter K of the classical k-NN model is equal to 1, etc.

From Table 8, we can see that, for the text and image datasets, the MMA-MF model achieves the best performance compared to other models. For the hybrid datasets 1 and 2, the fusion model designed by the MMA-MF model achieves the best performance compared to the classical k-NN, enhanced k-NN, SVM and random forest models. Overall, the MMA-MF model achieves the best performance in the four datasets.

**Table 8.** Comparison of experimental results on different models.

| Dataset Type | Model | Accuracy | Recall | F1-Score | Precision |
|---|---|---|---|---|---|
| Text | Char-CNN | 0.9563 | - | - | - |
| | BiLSTM | 0.9640 | - | - | - |
| | Naive Bayes | 0.9600 | - | 0.9600 | - |
| | ICRM | 0.9400 | - | 0.9500 | - |
| | **MMA − MF** | **0.9848** | **0.9796** | **0.9743** | **0.9850** |
| Image | Naive Bayes | 0.8504 | - | 0.9103 | - |
| | ID3 Decision Tree | 0.8900 | - | 0.9000 | - |
| | **MMA − MF** | **0.9264** | **0.9263** | **0.9263** | **0.9170** |
| Hybrid Dataset 1 | SVM | 0.9825 | 0.9825 | 0.9825 | 0.9810 |
| | Classical k-NN | 0.9783 | 0.9781 | 0.9783 | 0.9770 |
| | Enhanced k-NN | 0.9825 | 0.9830 | 0.9825 | 0.9820 |
| | Random Forest | 0.9833 | 0.9833 | 0.9833 | 0.9820 |
| | **MMA − MF** | **0.9842** | **0.9845** | **0.9841** | **0.9840** |
| Hybrid Dataset 2 | SVM | 0.9842 | 0.9844 | 0.9841 | 0.9830 |
| | Classical k-NN | 0.9808 | 0.9813 | 0.9808 | 0.9790 |
| | Enhanced k-NN | 0.9842 | 0.9844 | 0.9842 | 0.9820 |
| | Random Forest | 0.9833 | 0.9834 | 0.9833 | 0.9810 |
| | **MMA − MF** | **0.9846** | **0.9852** | **0.9845** | **0.9830** |

An ROC (receiver operating characteristic) chart, which can be considered as the average value of the sensitivity for a test over all possible values of specificity or vice versa. The ROC chart has the largest area under the curve (AUC) and is considered to be optimal. In general, an AUC of 0.5 indicates no discrimination (i.e., ability to diagnose whether an email is spam based on the testing), 0.7 to 0.8 is considered acceptable, 0.8 to 0.9 is considered excellent, and more than 0.9 is considered outstanding. Thus, we also use an ROC chart to further illustrate the performance of the MMA-MF model. We give four figures, Figure 4 shows the ROC chart for the text dataset and Figure 5 shows the ROC chart for the image dataset. Figure 6 shows the ROC chart for the hybrid dataset 1, for which the number of text dataset is equal to the number of image dataset. Figure 7 shows the ROC chart for the hybrid

dataset 2. From the above four figures, we can clearly see that the AUC indicators of the model on the four datasets are all greater than 0.93, which indicates that the performance of MMA-MF on the four datasets is outstanding for the spam detection.

In conclusion, it shows that the performance of the MMA-MF model is indeed better than other models that we mentioned, and the dropout technology does enable the MMA-MF model to be able to deal with hybrid spam, but also filter spam with only text or image data.

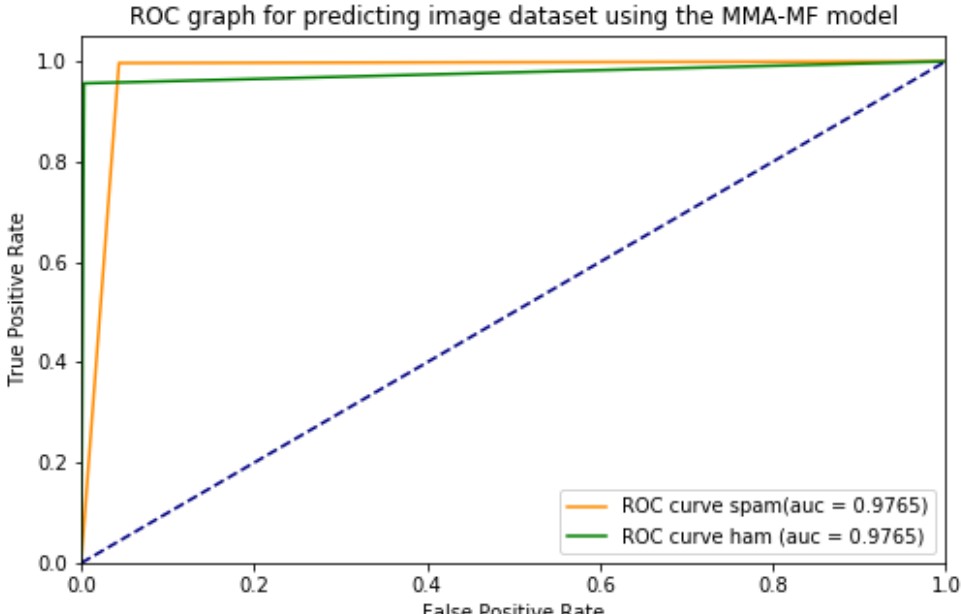

**Figure 4.** ROC chart for Text Dataset.

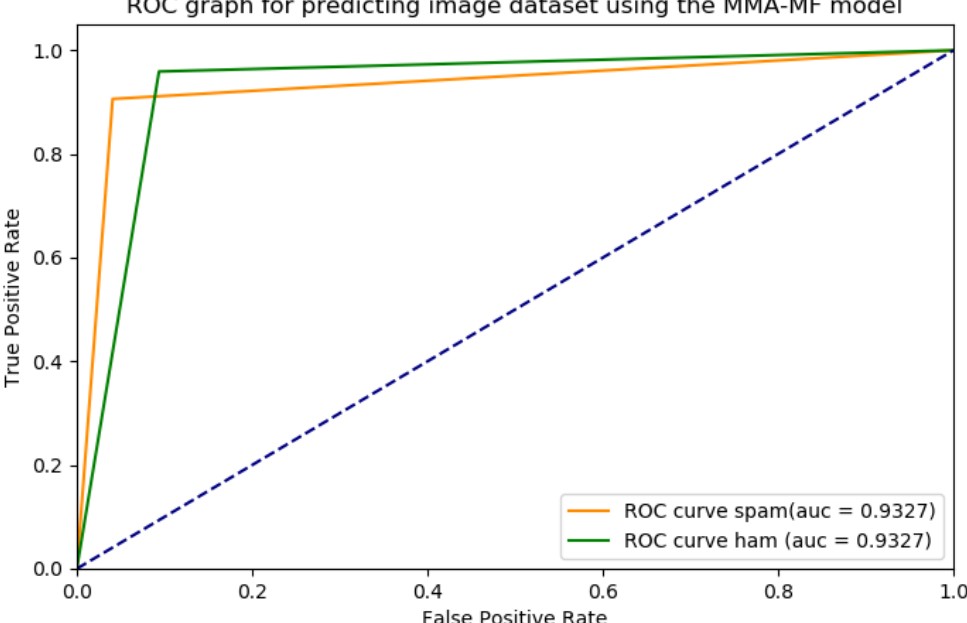

**Figure 5.** ROC chart for Image Dataset.

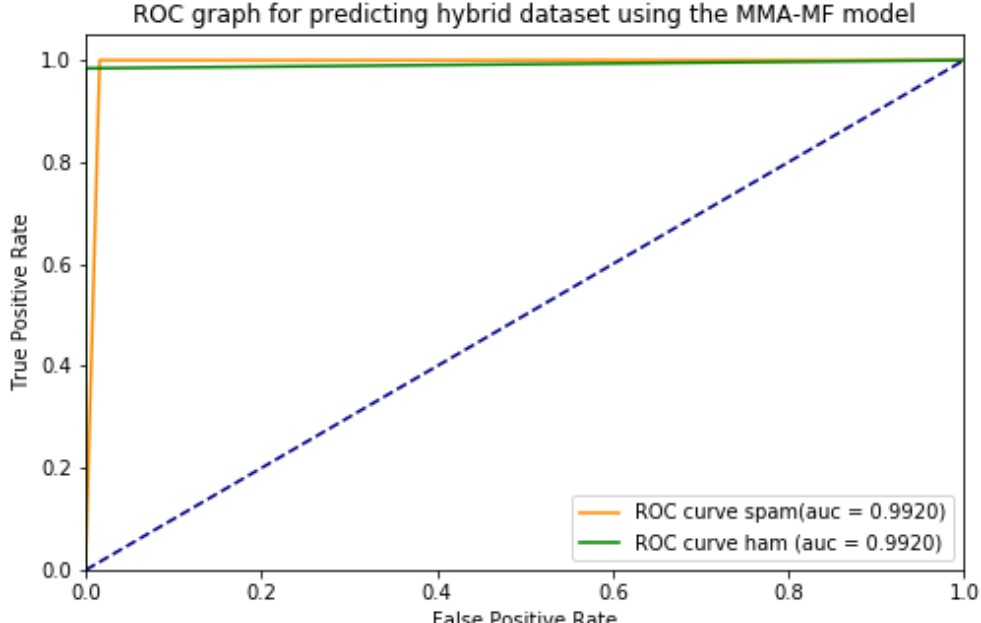

**Figure 6.** ROC chart for Hybrid Dataset 1.

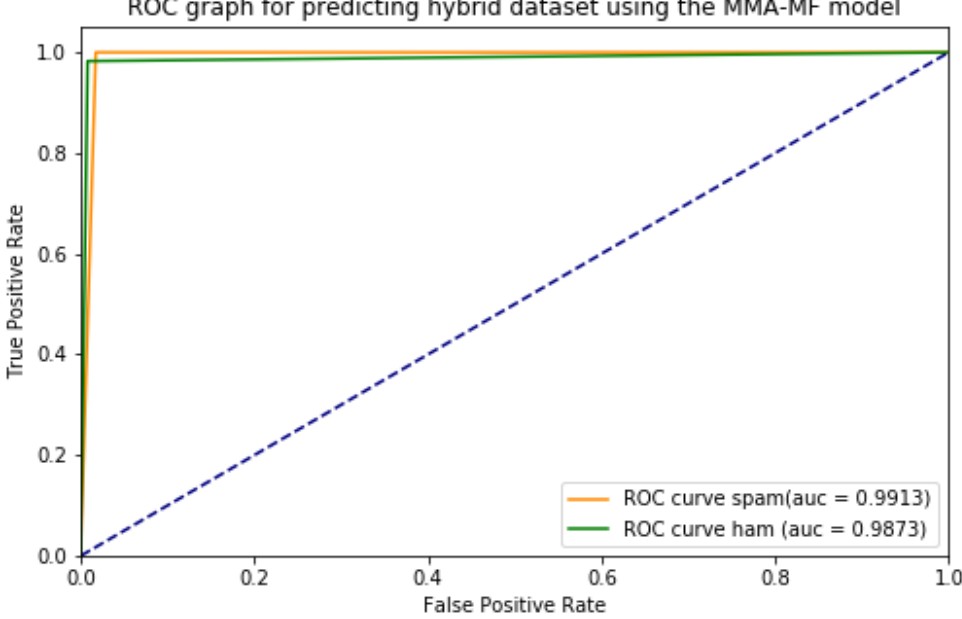

**Figure 7.** ROC chart for Hybrid Dataset 2.

## 5. Conclusions

We mainly introduce the multi-modal fusion architecture based on model fusion, which we called MMF-MF. The model combines the Convolutional Neural Network (CNN), Long Short-Term Memory (LSTM) network and fuses the two models by the logistic regression method to implement spam detection in a variety of email formats to improve spam detection rate. The advantage of the model is that it can not only filter hybrid spam, but also filter spam with only text data or image data, while other models can only handle text-based spam or image-based spam.

However, we have two issues that need to be solved in the future work. (1) From Table 5, there is no imbalance in our experimental dataset. However, in practical applications, spam detection datasets have a large discrepancy between the number of spam emails and non-spam emails. The solutions like one-class classification, few-shot learning and generative adversarial network methods should be

proposed to solve the imbalance between the positive and negative samples in the training dataset; (2) Owing to the fact that there is no real mixed email dataset for public use, the mixed email dataset is collected by splicing.

In the future, we hope to use the new technique just like the one-class classification method and a few-shot learning method to solve the problem of discrepancy between the number of spam emails and non-spam emails, and we will continue to collect more realistic mixed email datasets to improve the network structure of our model so that the model can get better spam detection performance.

**Author Contributions:** Writing—original draft preparation, H.Y.; writing—review and editing, Q.L.; supervision, S.Z.; project administration, H.Y. and Y.L.

**Funding:** This research was funded by the Sichuan Science and Technology Program (Grant No. 2017GZDZX0002, No. 2018GZDZX0006 and No. 2018FZ0097).

**Acknowledgments:** We thank William Cohen for providing the Enron dataset, and we would also like to thank Mark Dredze for providing the image dataset.

**Conflicts of Interest:** The authors declare no conflict of interest.

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
