# Peer review of "A Spam Filtering Method Based on Multi-Modal Fusion"

_applsci, doi:10.3390/app9061152_

Round 1

Reviewer 1 Report

1. k-fold cross-validation should be used to determine performance metrics 

2. Please include detail on how to find hyperparameters used in the model and optimizer.

3. The author should compare the present work with other spam detection techniques such as tweets, posts etc..

4.  Spam detection datasets usually have a large discrepancy between the number of spam emails and non-spam emails. (this is typical because it's very difficult to collect data on legitimate emails which are related/ closely resemble spam). Hence, solutions like one-class classification/ anomaly detection/ few-shot learning etc. are employed since there aren't sufficient negative samples to properly train a model. Authors can mention and discuss this problem in this paper.

Author Response

Dear editor,

          Thank you for your constructive suggestions. We have made corresponding modifications to our paper. Please read the word document for details.

Sincerely yours,

Hong Yang

Reviewer 2 Report

The way to create the mixed data is wrong since the mixed data contains information from training sets of the text and image data. To resolve this issue, the mixed data needs to be created by data which will not be used for the text and image datasets.

Besides using Logistic at the last step, please also use the following classifiers and report the performance: k-NN, an enhanced k-NN algorithm (DOI: 10.1109/THMS.2015.2453203), and SVM or random forest. The hyper-parameters for each of those classifiers need to be chosen carefully (using GridSearch).
A major revision is needed for this paper.

Author Response

(The authors gave the same response as above.)

Round 2

Reviewer 2 Report

A bit confused with the response and experiments in the revised manuscript.

What is the k-NN in the experiments? The classical one or the enhanced k-NN (CDNN)? You should report results from using both versions (classical k-NN and enhanced k-NN) and compare with yours and other methods. Including more classification algorithms in your experiments is better.

Author Response

Dear editor,

            Thanks to your feedback. We have made corresponding modifications to our paper. Please read the word document for details.

Sincerely yours,

Hong Yang
